# Health-Supportive Office Design—It Is Chafing Somewhere: Where and Why?

**Christina Bodin Danielsson** [1,*] and **Sara Hoy** [2]

1    School of Architecture & Build Environment, Royal Institute of Technology (KTH), 114 28 Stockholm, Sweden
2    Swedish School of Sport and Health Sciences, GIH, 114 33 Stockholm, Sweden
*    Correspondence: christina.bodin.danielsson@arch.kth.se or christina.bodin.danielsson@bredband.net

**Abstract:** This explorative case study investigates health-promoting office design from an experience and meaning-making perspective in an activity-based flex-office (A-FO) in a headquarter building. This small case study (*n* = 11) builds using qualitative data (walk-through and focus group interviews). A reflexive thematic analysis (RTA) of the experience of design approach was performed on this from a health and sustainability perspective, including the physical, mental, and social dimensions of health defined by WHO. Results show a wide range in participants' experiences and meaning-making of the health-promoting office design of their office building. The control aspect plays a central role in participants' experiences, including factors such as surveillance and obeyance, related to status and power, in turn associated with experiences of pleasantness, symbolism, and inclusiveness. Three main themes are identified in participants' experiences: (1) comfort–non-comfort, (2) outsider–insider, and (3) symbolism. The major finding of the study is the ambiguity among participants about the health-supportive office design of the office building per se and its various environments. There is a sense that it is chafing, due to dissonance between the intention of the office and the applied design.

**Keywords:** sustainable office; symbolic design; health-promoting design; dimensions of health; meaning-making; walk-through interviews; meaning-making; reflexive thematic analysis (RTA)

## 1. Introduction

We spend about 40% of waking hours at work, which, for most, is an office [1,2]. Hence, the work environment is important not only for employee health and wellbeing, but also for organizations and society [3,4], affecting employee health through acoustic and visual disturbances [5,6], infection risks [7], and sedentary work activities [8]. Likewise, the office design and individual environmental factors contribute to the psychosocial work environment, e.g., job satisfaction [9], leadership [10], and workplace conflicts [11].

In addition to the role as a marketing tool for organizations, the workplace can be used to foster innovation and creativity that is dependent on factors such as solving problems and development, e.g., [12]. It is important that employees at different levels in organizations can contribute, which the architectural design can facilitate [13,14]. Decision-making and group collaboration are especially affected by office design [15]. Interaction is central for information and knowledge transfer and social cohesion, affecting social networks and innovation [16,17]. The former study describes how more closed environments inhibited software engineers' communication, affecting their ability to work productively, either individually or as a team (p. 14 [16]). Office architecture can both facilitate or hinder interactions through various features, e.g., the plan layout, location, and gathering places [18,19]. In the latter study this is exemplified by how one employee working in a cell-office with individual rooms along a corridor, lacks the possibility to see people without making an effort to see what is going on at the workplace (p. 73 [19]). In addition, space design and distances matter [20,21]. For example, the work of Allen [20] has found that face-to-face interaction declined dramatically beyond about 30 m in an office. An aesthetic

office impacts employees' views of management positively [22]. Moreover, it seems to improve creative productivity [23,24]. The former review study has found that certain physical elements of the work environment can contribute to creativity, e.g., elements such as indoor plants and flowers, calm and inspiring colors, window view, and daylight (Table 1, p. 14 [23]). However, openness can also have a negative impact, e.g., declined job satisfaction and motivation [25], and less focus, with negative impacts on collaboration and creativity [26,27].

To conclude, existing research indicates that office design may to be a tool for organizations to improve employees' wellbeing and health as well as boosting their creativity.

### 1.1. Model of Occupational Stress and Environmental Stress—Theories and Concepts

The job demands-resources model (JD-R model) explains strain as a response to imbalance between demands on the individual and the resources available to handle these [28,29]. The JD-R model comprises a range of working conditions into the analyses, including both negative and positive indicators and outcomes of employee wellbeing, based on the idea that certain risk and beneficial factors are associated with job stress. Risk factors are classified as either job demands or job resources. Job demands (physical, psychological, social, or organizational) are associated with physiological and psychological costs (work pressure, emotional demands), while job resources, functional in nature are applied toward achieving work goals, reducing job demands, and stimulating personal growth and development. Two types of resources impact the development of job strain and motivation—workplace and personal resources [30]. Workplace resources (physical and social components) are powerful mediators of employee wellbeing, e.g., engagement and loyalty. According to the JD-R model, undesirable work resources, such as a poor work environment (physically, psychosocially), negatively effects employee energy, motivation, performance and health, while positive resources buffer the effect of job demands linked to physical and psychosocial work environment, e.g., support from colleagues.

Various architectural and environmental psychological theories focusing on the concept of personal control and theories related to this are relevant in order to link the JD-R model to the office environment. People strive for personal control, the individual's ability to bring about good events and avoid bad events [31]. The theories relate personal control to health and wellbeing as a cornerstone for health promotion through its influence on stimuli and stress [32]. The stress-reducing properties of personal control depend upon the nature of the response and the context. Lack of personal control can increase stress [32–34] or depression [35], but also reduce organizational engagement and innovation [36], while strong personal control increases job satisfaction [37].

Control at work is a multi-dimensional measure that includes task control, decision control, resource control, and control over the physical environment [37] and is an inevitable correlate of an organization [38]. Work organizations are an ordered arrangement of individual human interactions, which implies hierarchy associated with power and status. To convey cues about the status of an individual or group, symbolic design features are used [39]. This is symbolic identification [40] where the 'message' sent may involve the amount of space, nature of objects in an office and architectural design. Research about personal control in office environments is not all consistent. Nevertheless, control over the office environment appears to be positive for employee communication, environmental satisfaction, and perceived performance [41]. Personal control relates to privacy (visual and/or acoustic) and in its absence from distraction, as shown in the model by Lee and Brand [42] of the effect of personal control/distractions in the office environment on individual effort and group work. They found personal control over office environment to be positive for job satisfaction and group cohesiveness [42]. Another study has found personal control of ergonomics and ergonomics training in offices to improve environmental satisfaction and communication, but not effect psychological stress [43].

Focused on office design, we link the JD-R model to the office environment by here classifying it as a workplace resource, that when well-designed can be a powerful mediators

of employee wellbeing, but with the opposite effect when poorly designed. Environmental psychological theory describes the built environment's effect on human wellbeing as a result of architectural design and environmental factors, with stimulating or stressing influences [44]. Where a well-designed office environment enables focus and privacy when needed, when poorly designed it leads to stress, fatigue, and performance decline, [6,45]. The JD-R model suggests the office environment can also be a social resource enabling interaction and collaboration. According to the JD-R mode, the office environment can also be a social resource enabling interaction and collaboration [46], but also negative effects, such as tension and social conflicts [11]. As a workplace resource it can support or impair employee wellbeing. In line with the latter, non-stimulating and featureless environments, as well as being boring, are negative from the perspective of emotional health. Dubos [47] suggests people "can become fully expressed only when the (physical) environment provides a wide variety of experiences" (p. 339). On the other hand, stimulating environments are regarded as workplace resources, which, in accord with the theories presented, are visually interesting (i.e., complex, both spatially and ornamentally) [48,49]. Spaces that offer extended views and use natural material are also perceived as stimulating [24]. Likewise, the restorative qualities of nature [50] can also replenishing employee wellbeing and cognitive capacity. Mitchell McCoy and Evans [24] suggest similar effects by contact with nature even only via views of natural environments or exposure in interiors.

### 1.2. The Health Promoting Office

The importance of employee health for the welfare of organizations is well recognized. Healthy organizations prosper from a physical and psychological healthy workforce [51]. In recent years, organizations more actively promote health and wellbeing, and this is reflected not only in occupational safety and health (OSH) regulation in work environment legislation, e.g., the Swedish Work Environment Act, but also a part of corporate social responsibility (CSR), where organizations are expected to care for their employees (internal stakeholders), but also their local community and society at large [52]. According to the WHO, health consists of three interrelated dimensions: physical, mental, and social health [53]. Physical health is the wellbeing of the body and its functions [54], while mental health is a state of wellbeing in which the person can cope with the normal stresses of life and work productively and fruitfully, according to WHO [55]. Human interaction and contact is fundamental for mental health. It often concerns "problems in relation to interaction and communication" (p. 9 [55]). Mental health is a growing problem in modern working life, according to the organization, which is partly due to the greater importance of informal teams in the service economy than in the earlier industrial economy [55]. Social health is the core of our overall health, as humans need interaction with others [56], and it is central to an individual's quality of life, social efficiency and social achievement [57]. Social health with its five domains (social integration, social acceptance, social contribution, social actualization, and social coherence) reflects and evaluates our relation and functionality to people and the surrounding society [58].

Architecture can be used by organizations to promote employee health and wellbeing [59], and to improve employee health and productivity as part of a strategic branding plan to position the organization on the market. To be unique and to stand out is important in competitive markets and the office architecture is one strategy to cultivate the organization's distinctive character. This architectural branding is a materialization of brand values that can have external or internal focus in the design approach. The latter focus is also called employee branding [46], that despite its name, is often used as a recruitment tool. A well-developed example of a health-supportive design approach is the headquarter building of Medibank, a health insurance company in Melbourne, Australia. With its new headquarter reflecting the company's ethos [60], it aspires to become one of the healthiest workplaces in the world. The façades are covered by plants and the Medibank Place is designed around four themes (health, collaboration, innovation, and inspiration) to promote employees' physical, mental, and social health. Moreover, an activity-based workplace

design provides different work settings to choose between, to promote employees' health, ranging from quiet spaces to collaborative hubs and Wi-Fi-enabled balconies. The interior health design offers lighting mimicking natural daylight that supports the biorhythms and indoor plants and green walls. The green interior and exterior design is aesthetic, but is also an opportunity for restoration and stress relief in an urban working life e.g., [61–63]. The building's health design incorporates facilities to promote a healthy lifestyle, e.g., multi-purpose sports courts, an edible garden, and a demonstration kitchen for healthy cooking. In addition, convenient staircases encourage employees to take the stairs, in line with research that regards architecture a tool for physical activity [64,65]. Consistent with the Corporate Social Responsibility movement [53], so called CSR, the Midibank Place invites the surrounding community to use the building's amphitheater, cafes, shops, and a public park [61]. Support of social responsibility is found in architectural research on its impact on a sense of community and place attachment [66,67]; so-called social wellbeing is positive for the social dimension of health, [59,68]. Despite an increased organizational interest in health-promoting office design, e.g., Midibank Place, there is little research about this.

To our knowledge, there are no studies on health-promoting offices that investigate employees' perception and the role of different design features used. Hence, our study aims to do so, but being an exploratory study, it should only be seen as a first step in investigating the matter.

*1.3. Aim*

This explorative study aims to investigate and further understand user-experience of a health promoting office. For this purpose, we utilize an office with A-FO design investigating our research questions: How does employees experience and make meaning of a health-supportive office design? What role does different design features of an activity-based flex-office (A-FO) have for this?

## 2. Materials and Methods

This explorative study applies a case study approach. Doing so, it used a qualitative design (walk-through and focus group interviews) to explore the user-experience of a health-promoting office. One organization is investigated using interviews conducted at the end of 2019. Walk-through interview methodology is commonly used in the field of architecture to understand environments from a user perspective through a guided dialogue between lay people and professionals (i.e., employees and researchers in this study). A group of employees gather for a joint walk in the physical environment to understand and analyze it [69]. Afterwards, a focus group interview is conducted in which employees are encouraged to explore individual, shared and opposing perspectives and experiences [70].

The study was part of a larger project about the implementation of the activity-based flex office (A-FO). It has been approved by the Regional Ethical Review Board in Stockholm (No. 2018/1805-31/5). All participants provided their written informed consent.

*2.1. Our Case: Description of the Office*

In this case study the headquarter building of a large retail organization was investigated, built to be a modern, sustainable office certified as a BREEAM Excellence Building. It was tailored for the company based to provide the best conditions for innovation, collaboration, productivity, and wellbeing. All operations were gathered in one office building. Previously, people worked in different buildings made of traditional open plan offices with personal sit-stand desk workstations and bookable meeting rooms. The new headquarter consists of seven stories (31,000 m$^2$) with an A-FO design where about 2000 people share 5426 workstations, all with sit-stand desks, including meeting rooms, and smaller rooms for focus work, etc. Depending on task, one chooses between three different categories of workspaces classified as: dynamic, calm, or quiet work zones. The building is designed to support sustainability and health, which partly motivated the A-FO design. The health pro-

file was actively promoted internally, e.g., by a "sustainable week". Located in a suburban area with good public communications and all type of services; employees are encouraged to cycle by reduced car parking and increased cycle storage. Another health-supportive feature is the recreational floor at the building's top, with zones for exercise-work combined, recreation and team building activities. The building organized around three courtyards let in natural lighting from glazed ceilings.

Investigated Floor Types

Our study investigated three floor types in the building—entrance floor, office floor, and recreational floor.

*Entrance floor* (1st floor)—The entrance floor has two functions—a more public entrance zone and an internal organizational zone with dining areas and supplementary meeting and conference rooms. The entrance zone includes a reception area with lockers and café area for visitors to wait. To enter the internal company's side you pass a barrier, only for employees or visitors with permission. Here is a major dining area paired with a gathering/stage area, and two additional areas: a small and guest dining area. At the corner of the floor, by the major dining area is a lounge area aside the dining areas with supplementary rooms.

*Office floors* (2nd, 5th floor)—The regular office floors consist of five building sections organized around three courtyards and a middle section with the main staircase. By the office floors' entrance is a node of coffee/kitchen area located. The floors consist of several work zones in open spaces classified by the company into three definitions of atmosphere/activity (one divided into two sub-groups): (1a) Dynamic/home base area, (1b) Dynamic area, (2) Calm area (quiet but not complete silence), and (3) Silent area. We did both walk-through and focus groups interviews, but the former was not possible in the Silent area. Thus, interviews focused on four stop points located in work zones within the three former workspaces—the majority of work zones on the floor. Our four stop point areas were: (1) a so-called 'collaborative work zone' classified as dynamic/home base area; (2) a smaller work zone with project areas, next to zone one and classified as dynamic; (3) a large, bullpen work zone classified as dynamic; and (4) a calm work zone with back-up rooms classified as calm, but not complete silence.

*Recreational floor* (6th floor)—Located at the top floor of the building with terraces the intention of the floor is to offer employees a pause and recovery. Thus, the floor holds spaces for physical activity, collaboration, or avoidance of unwanted stimuli, e.g., large terraces, and different type of zones fulfilling the floor intention. These are: exercise-work stations (bicycle/treadmill), zones for physical activities/recovery (yoga and back stretching, table tennis), and alternative work zones (focus project area by terrace, meeting room with lecture seating).

*2.2. Participants*

Participants were recruited through the organization's intranet with information about the study and what participation meant. If interested, employees were offered an opportunity to have the study presented and to sign up for participation. Movie theatre tickets were provided as an incentive for participation. The recommended number of walk-through interview participants per group is five, and three groups or more, which adds up to 15 participants in total [69]. To incorporate as many perspectives and experiences as possible, we aimed for diverse groups with employees from different departments and office floors. However, we got 18 dropouts (five cancelations of scheduled walk-through interviews due to not being able to participate, three did not answer, and ten could not attend any of the dates for these interviews). The final sample was 11 participants (for details see Table 1), which were divided in two groups. They came from different departments and used primary different floors.

**Table 1.** Descriptive statistics for interview participants (*n* = 11).

| Variable | | Persons (*n*) | Frequency (%) | Year (Range) |
|---|---|---|---|---|
| Gender | Women | 9 | (82%) | - |
| | Men | 2 | (18%) | - |
| Age | | | | 29–58 years |
| Education | Post-secondary education (>3 years) | 7 | (63%) | - |
| | Post-secondary education (<3 years) | 2 | (18%) | - |
| | High school or equivalent | 2 | (18%) | - |
| Tenure | | | | 3 months–>18 years |
| Previous experience of A-FO | | 1 | (10%) | |

*2.3. Procedure and Data Collection*

The walk-through participants were taken on a pre-planned walking tour (1 h) around the office with two research staff, focusing on pre-assigned stop points that captured the different characteristics of the different floor investigated. Participants received a folder with background questions about participants (age, gender, educational level, department, work role, tenure, and years of A-FO experience). This was followed by questions to reflect upon for each stop point with drawings of each floor. At each stop point, participants reflected for approximately 3–5 min on how they experience and used the area, what worked well and not in the specific space. The following questions were utilized: *(1) How does this area feel to you, and how do you use this specific area? (2) What works well with the area, and why? (3) What is the challenge with the place, and why?*

Reflections were documented in the folder and obtained by the researchers after the walk-through. After the walk-through interviews conducted with questionnaires, a focus group interview (1 h) followed where participants talked about their visit to the different work environments on the office- and recreational floor. Participants also talked about their experience of the entrance floor, not covered in the walk-throughs, using the same questions. They sat in a circle facing each other and were offered refreshments and snacks before the start of focus group interviews. The interviews were audiotaped and transcribed verbatim by a third person through a transcription consultant company. A research staff member proofread all transcripts and cross-checked 20% of the transcripts against the audio recordings.

*2.4. Analysis*

In this study, the focus group interviews were used for the analysis that was performed based on the reflexive thematic analysis (RTA) described by Braun and Clarke [71,72]. RTA consists of six phases: (1) familiarizing with the data; (2) generating initial codes; (3) searching for themes; (4) reviewing themes; (5) defining and naming themes; and (6) producing the report. RTA aims to analyze patterns of meaning across a dataset, where themes are the output derived from the coding through a circular process. This means going back and forth between the steps of analysis. RTA was chosen since its theoretical flexibility fit with the study design. Two researchers were involved in the analysis, where one performed the main analysis and one assisted in the initial step and participated in the discussion and progress of the five latter steps. The analysis honed in on the study aim: the A-FO design in relation to employee experiences, with the focus on the way the office design supported or hindered aspects of health and wellbeing. It is done with a perspective that incorporated all three health dimensions described by WHO [52]: (1) physical, (2) mental, and (3) social health. In brief, described as follows:

- Social health aspects (on individual and group level, i.e., between colleagues in the team and between departments);
- Mental health aspects (stress, concentration, rehabilitating/ recovery);



- Physical health aspects (physical activity during the workday, ergonomic dimensions of physical setting).

First, the transcripts were read twice to familiarize and critical engage with the data. Parts that seemed interesting or challenging were reread and comments were added. Second, a systematic semantic and latent coding process was conducted. Notes were taken throughout the process of coding to support the search for and generation of initial themes. They were iteratively developed into themes and main themes, refined, and named, to develop robust detailed and nuanced answers to the study's aim. Here, codes were clustered together to identify patterns in the data, then mapped together again and again, trying to capture the meanings of the codes and combined into themes and sub-themes. Identified themes were clustered into three over-arching main themes.

Lastly, results were documented (see Figure 1 for main themes, and Figure 2 for themes categorized by the three health dimensions at different floor types). However, documentation was initiated from the start and an integrated part of the analysis. Therefore, RTA could be considered as circular process rather than linear. An example of the analytic process with its phases in the current study is displayed in Table 2 (See Tables S1 and S2 for overview of sub-themes and themes the main themes are based on).

**Table 2.** Example of the analytic process.

| Transcript Extracts Abbreviated (ex.) | Codes (Examples) | | (a) Sub-Themes | | (b) Themes | | (c) Main Themes |
|---|---|---|---|---|---|---|---|
| *It is good, / . . . / You can hide. If you don't want to be visible in office. / . . . / Nobody knows you are here* [on recreational floor] | 1.<br>2.<br>3.<br>4. | Avoid disturbance<br>Off the track<br>Seek seclusion<br>Focus | 1.<br>2. | Hide-away<br>Away from the disturbance | 1.<br>2. | Seclusion<br>Pleasantness | Comfort-Non-comfort |
| *It is clinical or generic / . . . /standard work-place, both view and interior design./ . . . /screams "work" to me* | 1.<br>2.<br>3. | Standard office environment—boring<br>Non-stimulating<br>Strict work focus | 1.<br>2. | Generic<br>Unpersonal | 1. | Anonymous | Comfort-Non-comfort |
| *Old identification artefact / . . . / we had in former office, . . . feels like it is missing. / . . . /could have gotten a lot more out of it. Instead of a plain entrance,* | 1.<br>2. | Lack of identification with new office<br>Miss strong symbolic artefacts of history from old HQ-building | 1.<br>2.<br>3. | (No) Symbol of recognition<br>Lack of belongingness<br>Nostalgia | 1.<br>2.<br>3. | Symbolism identification<br>Identity<br>Employee branding | Symbolism-Branding-Identification |
| *it* [the yoga zone] *is a bit strange* (laugh) *place/ . . . / Cold, draughty, very bright & yoga in the fluorescent lighting,/ . . . / never seen anyone* [use the space]. | 1.<br>2.<br>3. | Impossible to use-only symbolic design without use<br>Sterile & cold<br>Not a place to stay | 1. | Theoretic concept—Non-functional | 1. | Disneyfication–Gimmic | Symbolism-Branding-Identification |
| *Work space feels inno-vative . . . cool people at that department* [i.e., work zone]. *Thus, 40 percent wear caps, so hats on here.* [It is] *a little cooler there . . .* | 1.<br>2.<br>3. | Good environment–daylight, spacious<br>Belongs to the cool, high status group<br>Coded belongingness–entrance ticket "caps" on | 1.<br>2.<br>3.<br>4. | Creative environm. for privileged<br>Coded belonging–status group<br>Accessible only for the privileged<br>Socially excluding | 1.<br>2.<br>3.<br>4.<br>5. | Accessibility<br>Territorial behavior<br>Status<br>Privilege<br>Insider-outsider | **Insider-Outsider** |
| *Once introduced to it* [by colleague]. */.../ check out this, the secret room* [sec-lude, lounge zone]. *No-body knows this and then my colleague showed it* | 1.<br>2.<br>3. | Socially excluding–only for the knowledgable<br>Hidden—aside the<br>Secret—not accessible | 1.<br>2.<br>3.<br>4. | Seclusion –pleasant<br>Hidden—secret place<br>Accessibility—non-accessible<br>Soc.excluding—for knowledgable | 1.<br>2.<br>3.<br>4. | Accessibility–<br>Seclusion—hidden<br>Secretiveness<br>Insider-outsider | **Insider-Outsider** |

Notes: Presents three theme levels: (a) Sub-themes = report detailed meaning to central organizing concepts, (b) Themes = captures central organizing concepts of the sub-themes, (c) Overarching main themes (in total three) = captures central ideas underpinning number of themes identified. Three identified overarching main themes are: (1) Comfort-Non-comfort (colored in light grey), (2) Symbolism (colored in medium grey), (3) Insider-Outsider (colored in dark grey).

### 3. Findings

Focused on the employees' experience and meaning-making of a health-supporting office, three main themes were identified: *Comfort–Non-Comfort, Outsider–Insider* and *Symbolism* (see Figure 1). The main themes (including themes and sub themes) are in the text presented separately for the three floor types analyzed, i.e., entrance floor, office floor and recreational floor. In the presentation the floor types are titled with the identified main themes (including themes and sub themes) for each floor type.

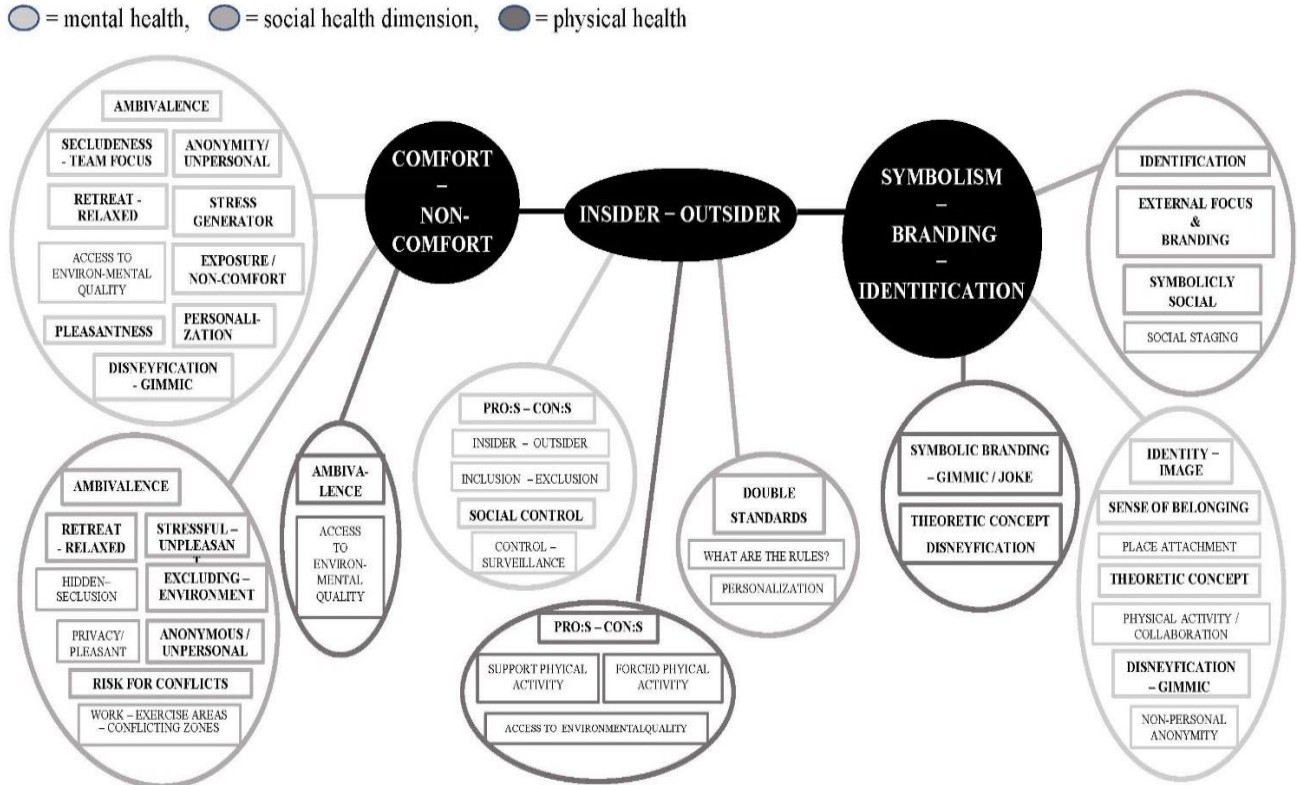

**Figure 1.** Thematic map—main themes including themes and some sub-themes. Notes: identified main themes (incl. themes and some sub-themes) are: comfort–non-comfort, insider–outsider and symbolism (branding–identification). Color scheme presents the health dimension which the included themes and sub-themes belong to.

The major findings in our analysis indicate a sense of lack of belonging and ambivalence in employee office experiences, reflected in the relation to the office environment apparent in the main themes. This manifests as a sense that something is chafing about the new head quarter building. This manifests as a sense that something is chafing about the new head quarter building. There is an ambivalence, although participants say they like the building's modernity, describing it as high-tech and 'fresh'. Despite these qualities, it is chafing somewhere, and a lack of belongingness and identification is further described. Various potential answers to the questions as to where and why can be sought in our findings below, and are ascribed to organizational arrangements, but also physical features. The latter includes different design features of office environment but also the choice of office type, i.e., the A-FO design, which, in turn, influences the psychosocial and spatial work conditions in the office.

This chafing sense participants describe about their new office building is reflected in the identified themes and sub-themes that are clustered into three main themes.

*3.1. Entrance Floor: Identification/Branding—Stressor Generator—Accessibility—Coping*

3.1.1. The Main Theme of Comfort—Non-Comfort

Exposure to two environmental stressors—crowding and noise—dominate participants' experiences on the entrance floor with negative impacts for emotional health. Uncontrollable exposure to environmental stress in areas overloaded with noise or visual distraction, e.g., the various dining areas, especially during lunchtime. Crowding occurs both in morning hours in the entrance zone and during lunch time in the dining area settings, with long queues to buy food, and a 'hunt' for seats, as diners with trays milling about the space. Participants note efforts for control over the mid-day challenges by shutting out environmental stimuli during lunch break. Discomfort due to stress of crowding in the dining area, makes it essentially non-accessible for employees sensitive to noise or visual stimuli, as the environment saps energy. Hence, coping strategies such as dining on their office floor or at off- lunch hours in the dining area are adopted by some employees. One participant describes it like this:

> *"Very noisy. I have a hearing impairment, so I have hearing aid, and that makes it even worse. I sometimes have a very hard time being able to communicate properly . . .*
> *"/Female participant, middle-aged*

The adaption of coping strategies due to environmental overload may come at the price of social wellbeing and social ostracization without full membership in the work community, hence, a lack of access to zones of environmental comfort for some employees', linking to the theme of insider–outsider. In addition, the periodically crowded and noisy grand entrance and reception zone is stressful, referred to as a congested subway entrance at rush hour with people passing gateways into the office. People queueing up by the long reception desk or calling for attention from inside the office add to its hectic atmosphere, although the lively climate energizes some participants.

In contrast to the former two high-activity areas, the secluded lounge area in the corner of the floor is hidden and unknown to many. The informed describe it as a secluded place of comfort and personal control—to withdraw to for private conversations and focused work. The soft and deep armchairs contribute to its relaxed atmosphere, a positive difference from the ordinary office workspace. One easily forgets time while working here, as described by one participant:

> *"Super nice and cozy environment . . . / . . . /people went home and Friday afternoon at three o'clock. I sat working there, then suddenly it was six o'clock. Man, I had forgotten to go home . . . . Because it was like this inviting* [atmosphere]. *It feels really nice."/*
>
> Female participant, older middle-aged

Associated to emotional wellbeing, this is a favorite zone in the building for those who use it. This is thanks to its seclusion, where secretiveness is as an important factor.

3.1.2. The Main Theme of Insider—Outsider

As a result of comfortable zones appearing as non-accessible or not, based on the employees' awareness or 'permissioned' access, reflect in participants' discussions aligns with the theme of Insider-Outsider. Zones meeting these criteria are the calm attractive dining areas (niches, guest dining area) and the secluded lounge area in a corner of the floor. Zones meeting these criteria are the calm, attractive dining areas (niches, guest dining area) and the secluded lounge area in a corner of the floor.

Lack of access concerning non-usability/functionality and the lack of inclusiveness of environments can be seen from two perspectives: firstly, from a design perspective, with environments not being accessible due to bad functionality, sometimes even a non-useable design, with reference to the design of the main dining area that causes environmental stress, making it inaccessible for some. This inhibits social gathering with colleagues and recovery during lunch, the midday break from work. Secondly, areas are described as non-accessible from a management perspective, due to inconsistent availability through

various booking systems, lack of internal promotion, and secretness. These problems concern calmer and more seclude areas of the floor (dining/meeting niches, the secluded guest dining area), that are booked in a different system than other spaces at the office and thus, are only bookable through the knowledgeable of the system. Lack of access is also a concern with regard to the secluded corner lounge area, a place not known to all. Those who do, describe it as a favorite place in the office building with a relaxed atmosphere; seclusion is a prerequisite for this. One participant describes the secrecy of the zone:

> "[I] *was once presented to it. Check out this secret room. Nobody knows of this. And then my colleague showed it* [the zone]*."*/Male participant, middle-aged

The zone was associated with emotional and social wellbeing as a consequence of environmental stress elsewhere, due to the inability to avoid unwanted environmental stimuli, i.e., stressors. In turn, lack of access for some employees affected their social cohesiveness, with potential social health consequences.

### 3.1.3. The Main Theme of Symbolism

Symbolism manifests on the entrance floor via two different aspects—branding and identification. Branding is here mainly external, i.e., with a visitor and customer focus. While identification instead references an internal focus on the organizational members, i.e., employees. Certain design features of zones on the floor ties to branding and identification symbolism, e.g., the entrance zone and the staircase seating area in the main dining area.

The symbolism embodied in the grand and spacious architecture of the entrance zone is designed for brand appreciation and pride. Although, for some, it is also a crowded and stressful place, described as less familiar and welcoming when compared to the old office building, with no relation to the organization. The critics lack a sense of context and history. This is reinforced by the loss of an iconic artifact from the former head office entrance, warmly spoken of by participants, as it promoted their nostalgia. Experiences of identification and sense of belonging have emotional implications that relate to place attachment, i.e., the affective bond between the individual and specific places (p. 274 [73]).

The other area where symbolism is reflected is in the staircase seating area in the main dining area, referred to as a symbol without nostalgic emotions. Instead, it was perceived as an attempt to relate to the company history. The external branding focus of the entrance floor is reflected in various design features perceived as mainly intended for visitors, e.g., the natural light and white color scheme, but also in the large digital screens, intended to signal professionalism and the grandeur of the company. The main external branding feature is the eye-catching staircase seating area, whose design and size signal a role of gathering place for employees. However, it is not used as such, as signs remind employees not to eat in the area, despite a lack of seating during lunch hours. One participant talks about it:

> "... *a mandatory identification feature, from the former office /* ... */ it has been implemented straight off as/* ... */ You mustn't bring food to the staircase. Would be nice to sit there, a bit on the side. There are signs that one cannot eat here. /* ... */ So, I do not know what to use it for, it is nice, but not many people us it* ... "/Female participant, younger middle-aged

The area's symbolic design is by some regarded as an attempt to evoke a sense of belonging, or a welcome, to the new office building; to others it is only as seating for larger presentations by the CEO and top management. Hence, its symbolic value from a social health perspective is without identification. The same applies to the guest dining room, not being intended for employees, but participants that knew of it expressed a desire to use it as is allowed, when not fully occupied. (As formerly described, this is unknown to many participants since it is booked through a separate system.)

Symbolism is associated with both emotional and social wellbeing, although less positively at the entrance floor. This is reflected in emotions, manifested in nostalgia for the former headquarters, as the new building appears disconnected to the participants. In

terms of social health, symbolism associates with this via its external focus on visitors in the symbolic design. This is described as theoretical, or not intended for use, hence without implication for employee social wellbeing, reflected at the staircase seating area in the main dining area that, as described by participants, instead of being a gathering place during lunch break, is as an aesthetic artefact, a focal point of orientation used for speeches.

### 3.2. Office Floor(s): Staged Collaboration—Accessibility—Coded Belongingness—Unclear Rules

### 3.2.1. The Main Theme of Comfort—Non-Comfort

The comfort varies in the different workspaces at the regular office floor(s) depending on variations in atmospheres. As such, there is a balancing act between stimulation and disturbance in the office spaces where a degree of privacy and the capacity for focus is important, even critical according to some, e.g., for those working with secretive assignments. There is a control aspect to participants' experience of comfort in shared workspace, as a corollary impact of social control and surveyance. The architectural features influence participants' experiences of workspace comfort, e.g., the positive impact from large windows, enabling daylight and exterior views. A view of greenery and sky associates with pleasantness, a view of roads the opposite. The varied and colorful interior is appreciated by participants. Despite the described architectural features, the one with most impact on comfort is the workspace size, as it correlates with visual/acoustic privacy and sense of exposure. This is reflected in the described comfort of different work zones, where the desirable smaller work zone contributes to a sense of 'safety', i.e., personal control. Other appreciated design features support individual work needs by facilitating the employee's personal control over the physical work experience. For example, control over environmental stimuli by providing calm areas when needed and ergonomic control with height adjusted workstations. Popularity of a work zone is determined by its environmental comfort, reflected in occupancy and personalization rate, leading to difficulties finding free workstations. The most popular zone on the office floor(s) is the smaller work zone with project areas, labelled as dynamic area by the organization. Described by participants as 'creativite', half the zone is inhabited by the marketing division, occupying most workstations, and the two project areas. In contrast, the most negatively described zone is the so-called 'collaborative' work zone by the entrance to the office floor (also labelled as dynamic). Participants describe uncontrollable exposure to people passing through, leading to acoustic and visual disturbance, and discomfort of exposure while working, causing focus and confidentiality problems. They portray difficulties in maintaining work capacity, emphasized by this high-performance organization. The importance of work zone size for experiences of comfort is demonstrated in the larger work zone's lack of popularity ascribed the lack of privacy, contributing to a sense of surveyance. The atmosphere is described as generic and non-personal, with words such as boring, grey, and sterile. Lack of aesthetic appreciation and non-stimulating design features also ascribes a crowded bullpen layout with workstation rows, and a boring view. The discomfort is amplified by the proximity to well-used conference rooms causing disturbances from passers-by and lack of access to back-up rooms in the zone as employees occupy these for entire workdays.

The ambivalence about the new office is most evident in the work zone classified as calm. Its near-silence rules and screens surrounding three sides of workstations dominating the environment adds to its enclosed and dull atmosphere. Opinions differ on how well the zone supports focus. Some are disturbed by the conversation, others like the incomplete silence, as this facilitates problem solving over the phone and decreases stress. Whether low-key talking is allowed in the zone is not clear among participants. The relative silence makes voices clearer, causing concentration difficulties. This zone, like others, has problems with back-up rooms occupied for extended periods but also the unclear rules on silence in the calm zone cause further tension and discomfort among employees.

A pattern in use of work zones exists relating to participants' experiences of comfort vs. non-comfort, associated with inter-personal relationships and social control affecting employee wellbeing, something apparent between zones described as boring or popular,

the boring zones are half full or fully vacant and the popular filled or marked by individual employees or employee groups. Social control prevails in both zones, exercised by the regular users, result in a sense of being overheard and observed by other employees. In the popular smaller work zone, an employee categorization is described, with clean-desk policy not applied for regular users, supporting their zone ownership. Sentiments about this are described in different forms, e.g., in laconic comments:

> *"My division is orderly; we follow rules and dare not to leave traces behind. We go 'by the book' while other groups leave traces. One asks oneself what the rules applied for our common environment?"* / Female participant, older middle-aged

An alienated, outside perspective is applied towards the popular smaller work zone by non-using participants describing it as a dynamic place 'where things happen,' attributed to the Post-it notes and things left on wallboards and desktops. Participants' ambivalence and outside perspective on the office environment's impact on social relations shows in the two zones with collaborative supportive design, i.e., the 'collaborative' work zone and the smaller popular work zone with project areas. The intention of the collaborative work zone by the floor's node, i.e., the entrance with a coffee area, is clear to all participants, but instead of being a place for collaborative and relaxed gathering, it is a place for quick, touchdown meetings due to its architectural design. Its plan layout and location cause both disturbances (acoustic, visual) and distractions from foot traffic in and across the zone as people converge upon the node of the floor. It is an environmentally stressful work zone inhibiting personal control or a relaxed atmosphere conducive to collaborative teamwork.

Ambivalence and an outsider perspective reflects the popular, smaller zone with project areas due to comfort problems ascribed to social control and territoriality. A subtle sense of unfairness transmits through participants, reflected as an ambivalence about the office. Notably counter-productive to both the A-FO design and the organization's intention to support collaboration in the new headquarters. Yet, social control is most evident in two other work zones on the office floor(s). These are: (a) the large work zone with workstation rows i.e., 'bullpen' office layout that despite classification as dynamic is not. Although conversation is allowed, it is described as uncomfortable, reinforced both by a sense of surveillance enabled by a broad overview of the space and by an anonymity, with workstation rows, and strict clean-desk policy. (b) The so-called 'calm' work zone, whose regular employees seek focus here and tend to guard' the silence by social control. This causes tension as no mutual agreement about silence rules exist in the zone and some employees work here as some phone talk is allowed.

Associated with both emotional and social wellbeing, variations in comfort experiences are described, reflected in participants' attitudes towards the different zones.

### 3.2.2. The Main Theme of Insider—Outsider

This theme appears in the employee's perception of workspace on the regular office floor(s) in various forms. It can be described as a mean of status. This is reflected by tension and the polarization of employees, associated with positive or negative experiences and attitudes towards the office environment. These can be categorized into pros and cons, affecting employees' emotional and social wellbeing related to work.

The theme concerns participants' sense of categorization of employees at the office reflected in how they read, i.e., understand, and talk about the office floor(s). An ambivalence towards the office is expressed as a matter of comfort and access to office environments that support collaboration and focused work, where the office environment is a tool to signal the type of employees and behaviors rewarded by the organization. Some amplified it and described it as used for social exclusion, thus categorizing employees into 'insiders' and 'outsiders'. Participants used various expressions for this, e.g., that one doesn't 'dare' to use certain workspaces, making such spaces off-limits to some. One participant talks about social exclusion:

*"People working there knows each other. / . . . / if you go there and try to find a worksta-
tion, it is almost like they look at you thinking: What are you doing here? Because people
have settled in there. It is a typical such a place where I would never even go and look for
a workstation there . . . "*/Female participant, middle-aged

Participants describe unfriendly looks from the 'owners' of a work zone towards newcomers who, having yet to 'learn' the unspoken rules, take a workstation here. The non-welcoming atmosphere results in some zones are never used by some employees, but regularly by others—hence less divisional interaction. That this territorial behavior is perceived as accepted by management, risks among 'the outsiders' lead to self-doubts. A sense of polarization is described cynically about the office as not being designed for ordinary nine to five working employees, but for youngers with an alternative 'Google work- attitude', for example by comments such as:

*"This is a team zone, I would never go here [to the zone] and work, I would feel strange.
The whole building, would have to be occupied for me to seek a workstation here (laughs)
I think. Don't want to disturb."*/Male participant, middle-aged

*"Caps on, and then you can sit here."*/Female participant, middle-aged

Thus, status and belongingness are in a sense coded, linked to certain attributes or manners, and participants' description of themselves and work in relation to the office. A participant at the logistics division claims the design of the small popular work zone with project areas does not suit them, since: "they are more square and their job very complex". This zone is described as tailored for the 'creative' marketing division occupying this zone.

This theme is associated with both emotional and social wellbeing as a consequence of its's role in self-identity and its association with others, in this context, employees' co-workers.

### 3.2.3. The Main Theme of Symbolism

The theme relates to the physical and social health dimension, with a focus on the latter. The office floor(s) are described from an outside perspective. The design approach is perceived as abstract, whose health benefits are foremost theoretical, where the office design is a Disneyfication, i.e., softening and 'funky' work environment, used to attract and retain employees [74].

The theme concerns the office floor's foremost focus on the supposedly social zones that are described by participants from an outsider perspective. Both the social and health-supportive purposes are clear to the participants, but the knowledge of management and the architect about collaborative works, i.e., how it works and what conditions it requires, is questioned. Their design is perceived as not grounded in reality, but rather a 'gimmick'. The criticism concerns an overly theoretical design approach to support physical and social health, which is not very comfortable. Participants were most outspoken regarding the 'collaborative' work zone, described as a symbolic stage for social interaction. The coffee lounge at the floor entrance is located at the central node on the office floor(s)—and is defined by a constant flow of people. The foot traffic emanates from two sources: those passing by or crossing the space heading to the smaller popular work zone behind this, and those visiting the personal cabinets placed here. Thus, neither its design nor location enable the intimate collaborative teamwork that brainstorming requires, due to multiple stressors such as noise and risks of being overheard and overseen. The stylish and colorful interior or large windows providing views and daylight do not seem to compensate for this, and nor does the clear collaborative supportive workstation design in the zone; although this is appreciated by some, according to others it is a non-functional symbols of social collaborative work. Thus, the symbolic design of this work zone divides participants. (For details see former Main theme of Insiders–Outsiders).

The symbolism of the supportive design is also described in terms of physical health, albeit to a lesser extent than social aspects. Participants expressed ambivalence about this, even if at a workstation level they appreciated physical health design features such as height

adjustability and dual computer screens, while the criticism concerned an overly theoretic symbolic design supportive of physical activity, describing it as a plain gesture, although certain details were appreciated, e.g., the meeting tables for standing up in the so-called cooperation zone. In fact, after the move to the new office with A-FO, some employees are more sedentary than before. This is credited to a workstation design enabling a free choice of work positions that reduces employee need to move around the office.

The symbolic design approach at the office floor(s), is associated with social and emotional wellbeing, but also to physical health to some extent.

*3.3. Recreational Floor: Pleasantness—Disneyfication/Theoretical Concept—Accessibility*
3.3.1. The Main Theme of Comfort—Non-Comfort

This theme was colored by the participants' ambivalence with regards to the recreational floor per se. While the design intention is appreciated, its support for comfort was in question, particularly regarding areas devoted for physical activity. Experiences were influenced by seclusion vs. exposure in different zones on the floor, but also by participants' perception of the floor as not welcoming for all employees' use. Hereby their comfort relates to all three dimensions of health—emotional, physical, and social health. Through the latter social dimension, the theme of comfort is also linked to the insider–outsider theme. Comfort experiences in different zones on the floor relate to architectural features such as: (a) design features such as windows and choice of material, and (b) plan layout, in turn associated with (c) zoning, i.e., the location of various functions in the plan layout.

Design features influence on ergonomic comfort is with regard to physical activity partly questioned, despite employee appreciation of the purpose of the recreational floor to promote physical activity. Participants had mixed feelings concerning emotional and ergonomic comfort in different zones on the floor, for example, the yoga zone (incl. back stretching) and the zone of the exercise-work stations. Again, their health-promotion purpose is appreciated, but their design features' support of ergonomic comfort questioned, e.g., the seating adjustments caused irritation and made some describe these stations as Disneyfications. This is how one participant describes it:

> *"I never use them just because of this [difficulty to adjust the seating positions]. I do not sit well on them. I cannot sit and read anything, or work something like that, at the same time."*/Female participant, middle-aged

Moreover, the ergonomic comfort of the meeting room with lecture seating awakes mixed reactions. Positive participants find the playful seating design comfortable, enabling different laptop working positions. While the alternative seating provokes some that regard it a waste of valuable meeting room space, being non-functional for brain storming. Another architectural feature such as its glazed walls making work sessions viewable from outside adds to the non-comfort. Despite this, the dominant architectural feature of plan layout and locations of functions affect comfort the most, through the ascribed perception of privacy, in turn influenced by the sense of exposure and personal control. This is most evident in the participants' experiences of two zones—the yoga/back stretching and the work-exercise station zones. The former zone has a cold atmosphere attributed to its sterile materials and cold temperature that, combined with its location in the major passageway to the terrace, does not promote a relaxed yoga vibe. The zone is intended for calm movement practices with wall bars and yoga mats, but functions as a passageway. Both location and architectural features counteract the purpose of the zone, inhibiting the relaxed atmosphere that these activities require. Another misplacement is that of the bike and treadmill workstation zones at the floor main entrance, whose exposure neither supports exercise nor work, and this is further reinforced by the mismatch between the zone of bike workstations and of table tennis. As well as discomfort, it causes self-consciousness, limiting both work focus and wellbeing. However, its view of treetops and sky also provides an opportunity for recovery.

One zone promoting comfort and wellbeing is the secluded project area located around the corner by the terrace. It is one of the most appreciated zones in the office building

thanks to two environmental qualities tied to its location. It enables (1) privacy by limiting disturbances and risks of being overheard without separation it from the recreation floor, facilitating the sense of community. It enables also (2) direct access to the terrace offering fresh air and quick leg stretching to promote recovery and invigorate work focus. A quality that according to some participants embodies the characteristics of an ideal physical work environment, reflected in comments such as this:

> *"used to work in a workplace where it did not exist, and could actually even dream a little about [having a terrace] how nice it would be to just be able to go out on a break and have a cup of coffee out in the fresh air."* /Female participant, middle-aged

As initially described, feelings are ambivalent regarding the recreational floor and comfort of various zones, e.g., the large terrace. Its intent for social breaks and outdoor work is appreciated, yet it is also associated with irritation and social friction due to poor wi-fi reception and people not picking up after themselves. Similarly, the comfort of the highly appreciated secluded project zone by the terrace is linked to ambivalent feelings. The desirable seclusion leads to an exclusivity, associated with social exclusion limiting the access of this due to different territorial behaviors. This includes e.g., personalization of this work zone and a sense that knowledge of this zone is protected, securing it for the own use, which associates comfort with the theme of Insider-Outsider. Attitudes to the most physical health-supportive zone on the floor—the table tennis zone—are also marked by ambivalence. Its support of physical health is described positively by all, being a 'healthier' and efficient method to recovery than a regular coffee break.

> *"If . . . you want to go there [to the recreational floor] /.../ see the view, walk a bit, relax. Then you hear that ping, pong. (laughter) /.../ So, it's not possible. And you just must leave the place. But it's fun for those who play table tennis."* /Female participant, younger middle-aged

Comfort experiences at the recreational floor are influenced by seclusion vs. exposure in different zones, and are associated with both emotional and social wellbeing, while physical health is associated with the floor per se and by the physical activity at various zones.

### 3.3.2. The Main Theme of Insider—Outsider

This is associated with the recreational floor per se, as it is not perceived as inviting to all employees. This is reinforced by limited internal marketing of the floor, but also lack of knowledge or difficulties to use and book various zones. The theme is also associated with specific zones with social, physical, and organizational aspects. Knowledge and use of different zones is interpreted as a sign of being an insider or an outsider, most obviously in term of accessibility to popular zones that are comfortable or supportive for wellbeing or work activity, and is apparent in the two very popular zones—the secluded zone for project work next to the terrace and the table tennis table zone, whose accessibility is tied to different territoriality behaviors. A sense that access to the attractive environment is administered by organizational conditions shines through, e.g., sanctioned personalization for certain employee groups, where those who know of appreciated zones, e.g., the seclude zone for project work, do not talk of it, and others do not even attempt to use it, as it is, anyhow, routinely occupied. This is how one participant describes it:

> *" a bit secluded like that, /.../ bright and spacious. I'd love to have a meeting there, but I did not even know the place existed. / . . . / could work efficiently here. / . . . /It is bookable as well, which I did not know / . . . / . . . the change is to find it."* /Female participant, younger middle-aged

Territorial behavior such as marking of ownership over popular zones, e.g., the seclude zone for project work, including leaving Post-it notes, writing on walls and on moveable screens. This personalization indicates a work session is on-going, but they are having a break. Whether this is sanctioned by management or not is unclear. Another territorial behavior—accessibility to zones—is perceived as coded by group affiliation and employee

status. This concerns the table tennis zone, where those with firsthand access, i.e., the insiders, includes employees perceived as 'creative' and working long hours, i.e., beyond 9–5 office hours. The less popular exercise-work stations are not coded with claims to ownership.

The theme is also manifested in polarized opinions about the floor, into the 'pros' and the 'cons', most evident with regards to the meeting room with lecture seating. The 'pros' include the opinion that it is a fun room for active and creative collaborative work by enabling different work positions, and also, that its wallpaper of trees adds to an attentive meeting atmosphere. The less positive participants, i.e., 'cons', describe the room as childish and useless for creative meetings, lacking functional seating, and walls for sharing and/or creating notes. Feelings about this room and the recreational floor reflect different attitudes toward the health-supportive office design, especially regarding physical activity. Some participants felt that employees were more physically active during the workday before the move to the new building, ascribing a health-supportive design approach not grounded in reality.

This theme is associated with all three dimensions of health. The polarizing nature of the theme is associated with both emotional and social health, as different sides of the same coin. It is also associated with physical health, due to its intention, with participants describing some zones as health promoting and others as too theoretical, even as gimmicks.

### 3.3.3. The Main Theme of Symbolism

This relates mainly to the recreational floor per se, but also to specific zones. Despite the dedication of the floor to the healthy organization concept, there was a skepticism of the applied symbolic design. This is most apparent with regard to the so-called yoga (incl. back stretching) zone, where neither the location nor architectural design facilitates the required relaxation. Thus, it is described as a gimmick, not based in reality. As such it is mocked and regarded only as a symbol of the concept of healthy organization:

> *"it [the yoga zone] is a bit strange (laughs) place /.../ do not really know how they thought (laughs). It is more or less a passage out to the terrace there. / . . . / Cold, draughty, super bright and yoga in the fluorescent lighting, / . . . / never seen anyone [use the space]. / . . . / Yes, rib chairs, good idea . . . / . . . / But it feels wrong."*/Female participant, middle-aged

Another clear symbol of physical health at the floor is at the table tennis zone. It is described from an outsider perspective as an managerial approach used by organizations to manifest themselves as a fun and youthful workplace. Participants' opinions differs about the zone. The positives appreciate it as a functional symbol that enables relaxion and fun activity, considering it a resource for recovery and a healthy alternative to a coffee break. The less positives describe it as an 'empty' symbol, without being anchored to the company identity and brand; very different from Google and other similar companies that use this type of Disneyfication. Another factor is that its exposed location reinforces the impression of it as mainly a symbol of physical activity, making some participants uncomfortable to play table tennis. This and the lack of tennis rackets inhibits spontaneous table tennis playing, according to its critics. A similar, but harsher, criticism concerns the 'exercise-work stations', described by participants as symbols without practical use. Again, the idea of combining physical activity with work is appreciated, though the health impact is doubted due to its bad ergonomics and exposed entrance location, which is not conducive to focused work. Symbolism associates positively with wellbeing at other zones, whose design fulfills employee emotional and work-related needs thanks to their symbolic and functionality design synchronicity. For example, the seclude project area by the terrace signals and enables focused teamwork, facilitating privacy and recovery. The large terrace that offers fresh air and a view both signals and enables work environment quality. This is associated with freedom and recovery, here described by one participant:

*" . . . it feels (laughs) good that you can go out and get some fresh air. I can feel a bit trapped sometimes in an office like this, towards the afternoon. / . . . / . . . it's good that the opportunity exists..."*/Female participant, younger middle-aged

In spite of regarding the terrace as a getaway for social gatherings, this symbol of a good work environment is questioned due to problems with poor Wi-Fi, and also noise and clutter. Feelings are mixed about the symbolic design of zones at the floor, e.g., the table tennis zone, where symbols of physical and social activity are associated with social exclusion and status (see former section Insider–Outsider). Some associate the zone with a conflicted, double meaning from the organization of both high work performance and having fun at work in Google-style, not easy given the workload. In addition, other zones evoke mixed feelings about their symbolism, e.g., the formerly described meeting room with lecture seating. Opinions differ; some appreciated the clear symbolism of the alternative seating and brainstorming teamwork. Others were annoyed, finding its symbolic design non-supportive and oversimplified, given the lack of regular meeting rooms. Moreover, a skepticism exists about how well the symbolic design at some zones supports physical activity. This concerns the 'exercise-work stations' whose symbolic design is described as non-functional and ergonomics, making it difficult to both write or read while exercising, but also changing seating positions difficult at the bike stations. Hence, the exercise-work stations are described as theoretic symbols of the idea of combining work with exercise. However, the clearest manifestation of a theoretical, non-functional symbolic design approach is that of the yoga and back stretching zone in the terrace passage, as it is not grounded in reality, according to participants. Despite the ambivalence toward the symbolic design of physical health-supportive zones, the participants like the floor as a symbol of a healthy organization, although its design approach is questioned, with more symbolic than functionally useful effective features. Ironically, the participants described the design as stimulating less physically activity during the workday than was experienced in the former office.

This was associated to all three dimensions of health symbolically by the floor per se offering a pause from office stressors and facilitating different health aspects. How well zones succeed vary, awakenings mixed feelings although their intentions are appreciated.

## 4. Discussion and Conclusions

### 4.1. Discussion

Our analysis of employee experiences and meaning-making of a health-supportive office building incorporated the emotional, physical, and social dimensions of health. In this discussion we debate the findings of our study, in association with the three health dimensions (see Figure 2).

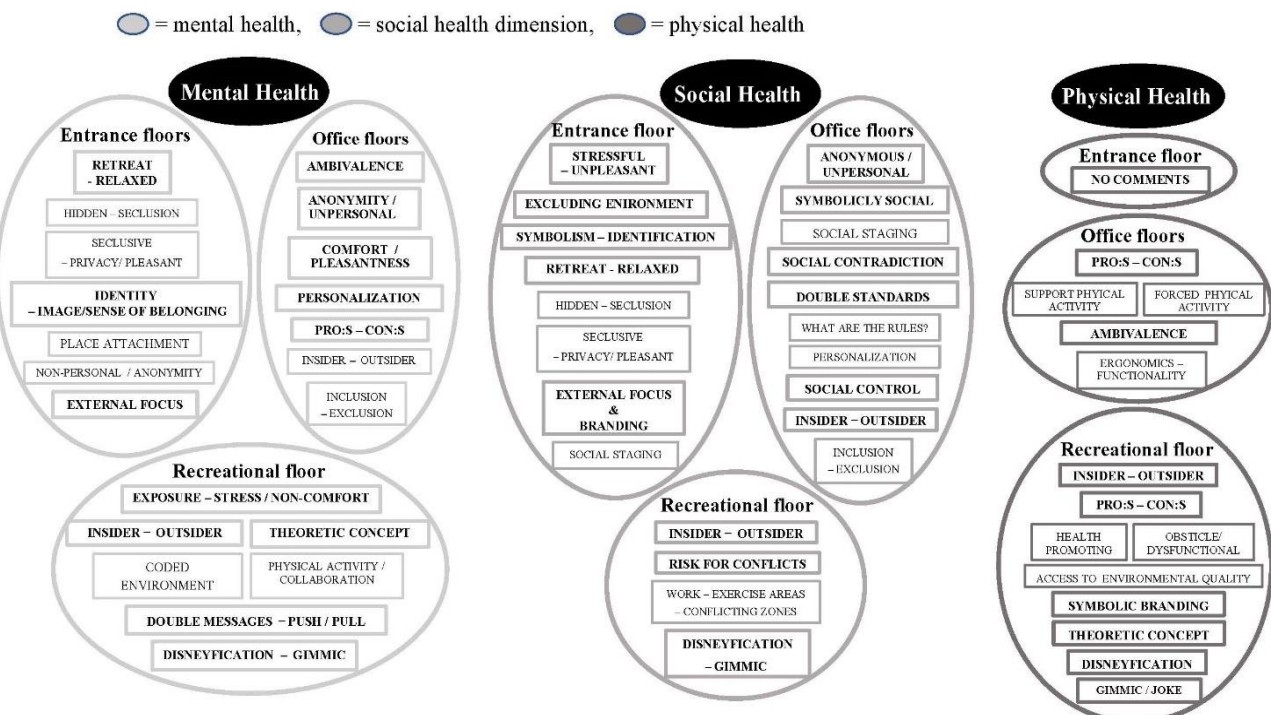

**Figure 2.** Thematic map—themes. Categorized within the three health dimensions at different floors. Notes: identified themes—level 2 (incl. some sub-themes) for the different office floor types categorized by the three health dimensions.

Employee experiences included both satisfaction and dissatisfaction with the case office building, but ultimately their experiences were colored by ambivalence, reflected in the three main themes as findings, with participants describing the building as more of an obstacle, neither supporting employee work or wellbeing and employees adopting different strategies to handle the physical environment. In this discussion, different interpretations of our findings from a health perspective to the formerly presented JD-R model linked to the office environment (see Section 1.1). Classifying this, a workplace resource that mediates employee wellbeing when well designed, when not the case, has the opposite effect. We use the three identified main themes as a framework for this discussion about findings in association with health dimensions.

The first main theme *Comfort–Non-Comfort*, associates at all floor types with emotional and social health, but only with physical health (ergonomics) at the recreational floor. Employee comfort experiences relate to personal control (physical, social) over the environment, often concerning seclusion vs. exposure in an environment. Emotional and social health, being different sides of the same coin, and their impact on comfort experiences is interrelated. Consequently, difficulties in avoiding exposure to environmental stress affect both aspects of employee wellbeing. This reflects in both individual focus and social interaction with implications from a comfort perspective, but also an outsider–insider perspective (see later discussion). At the entrance floor, comfort concerns control over environmental stressors such as noise, crowding and visual disturbances most evident in the dining areas during lunch, where lack of control over this, risks lead to stress symptoms related to discomfort. At the office floor(s) it concerns control over the work situation, that when not achieved generate stress and friction, causing discomfort (see Section 1.3). Comfort associates to emotional wellbeing via design features such as access to back-up rooms that support this, while a plan layout enabling disturbances inhibits it. It associates to social wellbeing via design features affecting comfort/discomfort, e.g., large open spaces enable overview but also surveillance, and even anonymity. The interplay between emotional and social health concerns tension/social friction over office rules such as disobeyance (e.g.,

occupying of back-up rooms) or unclarity (e.g., silence rules). At the recreational floor this interplay reflects in comfort problems due to privacy issues and unwanted exposure in different zones. Comfort impacts emotional health through the ability for relaxation and recovery, as well as focused work, while it impacts the social health by the ability for gathering and social interaction in the environment, influenced by design features, but also factors such as social inclusiveness. About physical health, comfort associates at the recreational floor to the design's support/hindering of physical activity and ergonomics, where poor environment causes both irritation and skepticism about the intent of health-supportive design and the applied design.

The second main theme *Outsider–Insider*, associates at all three floor types with emotional and social health, but with physical health only at the recreational floors that associates with all three health dimensions. The theme, by nature polarizing, concerns both the individual and the collective, i.e., the colleagues and the organization. It has a role for the self-identity in association to others, i.e., here office colleagues. This reflects in the experiences of all three floor types, where the specific environments are perceived as means to categorize employees by their internal status. Expressions of the outsider–insider theme vary between floor types. It associates to emotional and social health at all floor types as a consequence of environmental stress, where difficulties in finding acoustic and visual privacy affect both health dimensions. The stress and discomfort with health consequences such as stress symptoms (e.g., fatigue, difficulty focusing), and with social consequences such as reduced sense of community between employees. For employees sensitive to environmental stimuli, e.g., with hearing loss, adopting coping strategies such as less use of the entrance floor facilities or dining areas when crowded, may be most costly, as it may come at the price of less cohesion and sense of belonging with both colleagues, and the organization. The self-identity's association with others is reflected in the outsider–insider theme as a self-consciousness among some participants such as a sense of inherent unfairness about access to popular environments in the building. This causes tension with implications for relationships with colleagues, resulting in social friction that impact social health, most evident at the office floor(s) and recreational floor. Regarding the latter floor, polarizing effects are described about zones intended to support collaboration, social interaction, and physical activity as access to these are described as depend on employee internal status. At the recreational floor, experiences of outsider–insider was present in addition to the other health dimensions, also reflected in perception of various zones at this floor. Features intended to support physical activity polarize the participants, e.g., the table tennis zone that positives described as supportive of both cohesion and physical activity, while negatives see it as non-inclusive, regarding access to it a status marker.

The third, final main theme *Symbolism*, associates with health to various extents at the different floor types, but with all three dimensions only at the recreational floor. The theme manifests mainly by a nostalgia for the former headquarters, while the new is not described in these terms. This may be due to lack of place attachment, the affective bond between the individual and specific places (p. 274 [73]), that associates positively to emotional health. In our case, when participants talk of the new office workplace it does not align with their feelings, as they feel a lack of sense of belonging and identification with the office.

At the entrance floor, the staircase seating area by the main dining area is the main manifestation of symbolism. Intended to evoke a sense of belonging and welcome, it is associated with both emotional and social health. Despite this, it is mainly a place for CEO and top management presentation with an external brand focus short of identification value, according to participants, a result of both its theoretical design approach and not being intended as a gathering or eating place. Although perceived as aesthetic, this is source for irritation, due to lack of seating during lunch. The same external brand focus applies to the guest dining room, unknown to many due its separate booking system. Participants who did, found a positive symbolic value in it from an emotional perspective, attributing the exclusive and intimate atmosphere, separated from the stress outside. At the office floor(s) symbolism associates to employee social health, and partly to physical

health. The support of social health is symbolic without true value due to the theoretical design, not truly supportive of collaborative work. About the symbolism of the physical health support, the company's health strategy and choice of design approach are perceived positively. The symbolism of functional and ergonomic features such as height-adjustable workstations and dual screens are appreciated. Still, there is a gap between the symbol of physical activity and office work at departmental level due to an ill-considered A-FO design not based on daily office work. The recreational floor symbols health per se, manifesting recovery and pause from daily work stressors associated to all three health dimensions. The ability of symbolism at different zones depends on the how well they signal stimulation and seclusion, both central for emotional and social wellbeing. The zone with most positive symbolism at the floor and the building as a whole is the secluded zone for project work by the terrace, signaling both privacy and work focus, qualities further reinforced by the zone's direct access to outdoor fresh air. Another symbol of social and physical health is the table tennis zone. Meant to be a symbol of social togetherness, but to some it symbolizes exclusion, as access to this links to employee internal status. This causes ambivalence. So does the organization's contradictory message with the zone, promoting both employee work performance and their fun at work. Similarly, ambivalence about the exercise-work stations exists. The symbolism of combining work and exercise is appreciated, but not the theoretical design, described as a non-functioning joke.

Finally, when discussing the identified main themes associations with different aspects influencing health and wellbeing, we find all the themes associated with emotional and social health dimensions, while participants' describe experiences less associated to physical health dimensions. As we analyze our findings, we interpret these as that the employee office experiences in relation to health to a great extent can be ascribed to design features of the various zones investigated in our study. This being said, one needs to remember that doubts of the building design's direct health benefits such as stress reduction, recovery or physical activity, were expressed. For example, regarding physical health benefits, some participants even claimed people were less physically active in the new building than prior due to its design, despite contrary intentions. Then again, concluding this discussion on our findings associations with health dimensions, one has to bear in mind other potential explanations for participants' experiences may exist, although not in focus here.

### 4.1.1. Our Finding in Relation to Existing Research

To our knowledge there is no research that specifically investigates employee experiences of a health-supportive office design and the role of different physical workplace features for this, although, some attempts to summarize features central for office experiences have been done [75]. Research has found cell-office employees satisfied with acoustic and visual privacy tend to be more satisfied with various design features (Bodin Danielsson and Bodin, 2009), but also with ambient factors (temperature, ventilation, lighting), something the study suggests depends on the greater control over the physical workspace. Other office studies have found personal control over the physical workplace to influence for both employee job satisfaction [37] and workplace satisfaction [42]. The presented research is in line with the overall result of our exploratory study that highlights that office zones offering seclusion and privacy, i.e., personal control, are most popular among office employees, maybe even more in an A-FO workplace design such as our case study. The preference for these qualities is reflected in our finding that workspaces in corner locations, enabling overview and personal control at the office floor(s) always are occupied, a behavior apparent throughout the building, according to participants. The two favorite zones in the building—the lounge area on the entrance floor and the seclude project area zone at the recreational floor—are also defined by seclusion and privacy. The latter, supportive of collaborative work is a smaller open workspace for 6–10 people with large windows and direct terrace access. These design features of the latter zone are highly appreciated by participants. Other research has also found these supportive of workplace satisfaction, e.g., smaller open workspace size appears to be central for this in offices of A-FO design [76].

Good access to daylight and to an outside view increase office employee environmental satisfaction as well [77] This is also true for unique design features such as a terraces at office [22]. Moreover, the latter study found this positive for employee perception of the workplace. This is also in line with our findings, though it did not compensate for other negative office experiences. In addition, our exploratory results find support in studies on the importance of access to meeting rooms and back-up rooms in open workspaces for employee personal control and job satisfaction [42] and their satisfaction with workspace contribution of office designs well [78].

### 4.1.2. Strengths and Limitations

A strength of this case study is its walk-through design, that allows the participants an immediate experience of the setting that was asked about followed by the focus group interviews. Two researchers were present at both data collection and the analysis, and peer debriefing has been a recurring feature throughout, strengthening the study's credibility [79]. A clear description of the research process has been provided, adding to the dependability of the study. Despite the strengths of the study, a few limitations should be addressed. The most apparent concerns the sample size. This is due to difficulties with recruitment of participants resulting in only 11 respondents divided over two groups in our final sample. We had 18 dropouts and only two men were recruited. This means that the initial aim of a minimal of at least 15 participants divided over three focus-groups as is recommended by de Laval [70] was not reached. However, given that our study is exploratory by nature, i.e., a first step in a larger research intent, this is not a grave limitation. Moreover, Braun and Clarke [80], argue that 'how many' data items you have is secondary to the interpretive work, through analysis, that you perform and the meaning that is generated. Another limitation was a need to limit the time for briefing and focus group interviews from three to two hours, due to participants demand on it being shorter. This led to the data collection from the entrance floor being cut to a focus group discussion only, eliminating the initial work-through interview with participants.

### 4.2. Conclusions

To conclude, the results of this small, explorative case study of health-supportive design indicates it has implications for employee meaning making of their workplace from a health perspective, but also in other regards. The impact of this design may be ascribed to the design features of the office environment in combination with organizational arrangements. Despite a pride among employees for their new office building and its design, the major finding our study is the chafing feeling participants expressed as an ambivalence about its design and function. This ambivalence seems to lie between the intention of the supportive health design and employees' own experiences.

Focusing on experiences related to the three health dimensions in our analysis, we found the impact of the dimensions varying, with less impact on the dimension of physical health than emotional and social dimensions. However, we found most doubts about the design features' actual support of physical health. Some participants claimed people were less physically active in the new building due to its design. Ambivalence was also expressed about the use and purpose of the design. They were positive about the stations for movement, but their placement and designed made them non-functional, and also perceived as a gimmick, even a joke.

Support for the outspoken intent of the new office's design to unify employees, to encourage collaboration and innovation was in our findings only found at individual workstation level. Instead, participants described polarization between different employee groups, expressed as a sense of a difference in status between groups reflected in territorial behaviors in popular work zones whose environmental conditions offer focus, stimuli, or recover. The zone's location and zoning determined this, e.g., being next to an activity node or a mismatching activity cause problems manifested in the main theme of *Comfort–Non-Comfort*, influencing employee emotional and social wellbeing. These experiences are

influenced by other design features as well, such as spatial seclusion, size of workspace, and anonymous architecture, but also choice of office type. In this case study, the A-FO design that reinforces spatial conditions contributes to sense of lack of belonging, or being "lost in space". A striving for a personal corner in the office to enable both coherence to a place and the people working there is also described, including spatial, but also social and organizational dimensions, concerning accessibility and exclusion to certain environments manifested as experiences of polarization. These experiences are reflected in the main theme of insider–outsider, as well as a gap between the intent and employee experiences of the building, potentially depending on aspects such as a perceived focus on external branding and a non-reality-based design approach. Reflected in the main theme of symbolism, it is described as a too theoretical design approach, e.g., as a Disneyfication of the space. This is ridiculed by some participants, and by others interpreted as a lack of knowledge or interest in employees among management.

In summary, our analysis leaves us with a sense that the main office experience is one of ambivalence and that it is chafing somewhere between the idea of the design and its implementation, and supportive qualities while being used (or not). Beside the described ambivalence towards the office design, another major finding of our exploratory case study is highlighting the difficulty in designing a good workplace, a complex assignment due to the organizational context. Maybe even more so when applying an unusual design, such as a health-supportive design, and when the intent of this is outspoken, it is important the design is well founded and reality based. Despite problems with stress and environmental comfort, as well as experiences of exclusion and polarizations related to the new office, we found that participants were proud of the intention of their new office building and took pride in working for the company. Future research may consider further investigate health-supportive design, but also issues such as pride in relation to environmental satisfaction from an organizational perspective to identify what supports this.

**Supplementary Materials:** The following supporting information can be downloaded at: https://www.mdpi.com/article/10.3390/su141912504/s1.

**Author Contributions:** The study was performed by senior researcher (C.B.D.), architect and associate professor with a Ph.D. in Architecture (human-environment interaction) and research assistant/doctoral student (S.H.) with a BSc in Public Health and MSc in Sport Science. They both planned and executed the data collection as well as the initial analysis and study structure. C.B.D. performed the main analysis and S.H. assisted in the discussion and the five latter steps. C.B.D. did the writing, with the exception of the method section written by S.H. Both group members had earlier experience of conducting interviews. All authors have read and agreed to the published version of the manuscript.

**Funding:** This research was funded by the Knowledge Foundation (Swedish: KK-stiftelsen), grant no 20170116.

**Institutional Review Board Statement:** The study was approved by the Regional Ethical Review Board in Stockholm (No. 2018/1805-31/5).

**Informed Consent Statement:** All participates provided their written informed consent.

**Data Availability Statement:** Data available on request.

**Acknowledgments:** We thank Anne Richer, associate professor at Karolinska Institute, project leader of the larger research project that this study was part of.

**Conflicts of Interest:** The authors declare no conflict of interest.

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
