# Peer review of "Health-Supportive Office Design—It Is Chafing Somewhere: Where and Why?"

_sustainability, doi:10.3390/su141912504_

Round 1
Reviewer 1 Report
The study addresses an important and current issue: the assessment of the level of comfort and physical, emotional and social health in the workplace. Three main dimensions are analyzed: Comfort-non-comfort, Symbolism-Branding-Identification, Insider-Outsider. Although the methodological approach followed is consistent with previous studies, in my opinion, no elements of originality emerge from the work. The approach adopted, already corroborated in many other studies, does not seem to present any element of innovation. Furthermore, the selected reference sample (n = 11) may not fully represent the entire employee population. A general plan and the specific plans of the various levels of the building complex under study are missing, in my opinion fundamental for the reader to orient himself and have a complete perception of the spaces. The results of the survey are not accompanied by proposals for design solutions to improve the architectural spaces (what could be done to improve the situation?). In the text there are several typos to be eliminated, as shown in lines 48-49 and lines 176-183. The two introductory subparagraphs (1.1 and 1.2) could merge into a new paragraph on the "state of the art". The paragraph “Method discussion” could also be merged into “Discussion and Conclusion” (or become a sub-paragraph of it).
Author Response
We are happy to read that we have been invited to resubmit our article “Health supportive office design – It is chafing somewhere. Where & why?” with clarifications and changes as suggested by the comments from the reviewers. We summarize here changes we have made in the revised article as a response to reviewers requests, realizing we in our effort to keep the text down in the previous version made it difficult to understand our analysis and results. This includes for example, more concrete examples from both our own study and from referenced research.
Please see the attachment.

Reviewer 2 Report
This paper studies the employees workplace experience and meaning-making of the design approach in a case study of head office building. This was done with a reflexive thematic analysis incorporating the emotional, physical, and social dimensions of health. Results of RTA of participants experiences of our case office shows both satisfaction and dissatisfaction among participants, but ultimately an ambivalence. They appreciate the good intention of the building design but find it hard to read. Frustration exists where some participants describe adopted strategies to handle the office environment to be able to work. The results seem to be new and the topic is timely.
Decision-making and group collaboration are affected especially by office design. Interaction is central for information and knowledge transfer and social cohesion, affecting social networks and innovation. Some concrete examples would be useful.
In section 1.2, when dicussing mental health and welbeing, you may also want to refer to the following study. Online screening of X-System music playlists using an integrative wellbeing model informed by the theory of autopoiesis.
There are only 15 participants, which seem to be a quite low number. Have you considered using an hypothesis testing analysis to validate the results and and hypothesis?
The paragraph starting from line 303 is not very clearly written. More details and quantitative comparision is needed.
I would suggest the authors to use word cloud to illusrate the results if possible. The long narrative is probably not the most useful way to demonstrate the points.
The theme of Symbolism requires more elaboration. I am at a loss to understand many of the arguments in this part perhaps due to my lack of familiarity here. But more explanations should be beneficial.
I noticed multiple lines starting from line 929 are shaded in blue, which is suspicious.
Author Response

(The authors gave the same response as above.)

Reviewer 3 Report
Thank you for this opportunity to revise the manuscript titled "Health supportive office design – It is chafing somewhere. Where & why? " that was submitted to Sustainability.
The relevance of the research realized by the authors of the paper is obvious. I leave several comments about the manuscript which are listed below:
1. The sample that is used in the empirical part of the study is very small (N=11). This limits conclusions that can be drawn from the empirical investigation.
2. Limitation of the framework and suggestions for future studies as well as practical implications should be added, substantially.
3. The method should be rewritten in a more organized way.
4. Discussion should be expanded.
5. Unfortunately, this work is presented in such a way that its value is difficult to assess. It remains unclear on the basis of which the hypotheses were formulated.
I recommend that the authors seriously rework all the material so that it can be evaluated in the future.
Author Response

(The authors gave the same response as above.)

Round 2
Reviewer 1 Report
The manuscript has undergone a substantial revision. Many of the doubts that emerged from the previous version have been answered through arguments integrated into the text. The description of the environments has improved. Typos have been removed. The overall structure and legibility of the paper have improved overall.
Reviewer 2 Report
The revised paper is in good shape now. It can be accepted.
Reviewer 3 Report
Thank you for sending this manuscript for another round.
The authors did a great job in processing the comments.
I think the paper is in good shape and can, in my opinion, be published in Sustainability.